# Shaping Silver Nanoparticles’ Size through the Carrier Composition: Synthesis and Antimicrobial Activity

**DOI:** 10.3390/nano13101585

**Published:** 2023-05-09

**Authors:** Margherita Cacaci, Giacomo Biagiotti, Gianluca Toniolo, Martin Albino, Claudio Sangregorio, Mirko Severi, Maura Di Vito, Damiano Squitieri, Luca Contiero, Marco Paggi, Marcello Marelli, Stefano Cicchi, Francesca Bugli, Barbara Richichi

**Affiliations:** 1Dipartimento di Scienze Biotecnologiche di Base, Cliniche Intensivologiche e Perioperatorie, Università Cattolica del Sacro Cuore, 00168 Rome, Italy; margherita.cacaci@unicatt.it (M.C.); wdivit@gmail.com (M.D.V.); damiano.squitieri@unicatt.it (D.S.); 2Dipartimento di Scienze di Laboratorio e Infettivologiche, Fondazione Policlinico Universitario A. Gemelli IRCCS, 00168 Rome, Italy; 3Department of Chemistry “Ugo Schiff”, University of Firenze, Via della Lastruccia 13, 50019 Sesto Fiorentino, Italy; giacomo.biagiotti@unifi.it (G.B.); martin.albino@unifi.it (M.A.); stefano.cicchi@unifi.it (S.C.); 4Consorzio Interuniversitario Nazionale per la Scienza e Tecnologia dei Materiali (INSTM), 50121 Firenze, Italy; 5ICCOM CNR, Via Madonna del Piano 10, 50019 Sesto Fiorentino, Italy; csangregorio@iccom.cnr.it; 6Cromology Italia S.p.A., Z.I. Porcari, 55016 Lucca, Italy; luca.contiero@cromology.it; 7IMT School for Advanced Studies Lucca, Piazza San Francesco 19, 55100 Lucca, Italy; marco.paggi@imtlucca.it; 8CNR SCITEC-Istituto di Scienze e Tecnologie Chimiche “Giulio Natta”, Via Fantoli 16/15, 20138 Milano, Italy

**Keywords:** silver nanoparticles, graphene oxide, salicylic acid, titanium dioxide, antimicrobial resistance

## Abstract

The increasing resistance of bacteria to conventional antibiotics represents a severe global emergency for human health. The broad-spectrum antibacterial activity of silver has been known for a long time, and silver at the nanoscale shows enhanced antibacterial activity. This has prompted research into the development of silver-based nanomaterials for applications in clinical settings. In this work, the synthesis of three different silver nanoparticles (AgNPs) hybrids using both organic and inorganic supports with intrinsic antibacterial properties is described. The tuning of the AgNPs’ shape and size according to the type of bioactive support was also investigated. Specifically, the commercially available sulfated cellulose nanocrystal (CNC), the salicylic acid functionalized reduced graphene oxide (rGO-SA), and the commercially available titanium dioxide (TiO_2_) were chosen as organic (CNC, rGO-SA) and inorganic (TiO_2_) supports. Then, the antimicrobial activity of the AgNP composites was assessed on clinically relevant multi-drug-resistant bacteria and the fungus *Candida albicans*. The results show how the formation of Ag nanoparticles on the selected supports provides the resulting composite materials with an effective antibacterial activity.

## 1. Introduction

The antimicrobial activity of silver has been known for a long time, and presently silver is included, in different forms, in various products for medical and healthcare uses, food packaging, and clothing [1]. The advent of nanotechnologies led to the discovery that silver nanoparticles (AgNPs) show enhanced antibacterial activity, thereby focusing the attention of researchers and industries on this nanomaterial. In the last two decades the market for AgNPs has been growing speedily, and AgNPs have been combined with different products to provide bactericidal capacity [2,3].

Notably, AgNPs show antimicrobial activity against a variety of pathogenic microorganisms, including multi-drug-resistant bacteria, and do not induce the development of resistance [4,5,6]. This opens significant avenues in the fight against antimicrobial resistance. Accordingly, numerous efforts have been focused so far on the use of AgNPs to prevent infections caused by multi-resistant organisms and on the fabrication of new bactericidal products for decontamination or infection treatments [7,8]. The potential of AgNPs as antibiotics arises from the multiple mechanisms associated with their antibacterial activity (e.g., AgNPs disrupt the cell membrane, impair the respiratory pathways and intracellular components, interact with biomolecules bearing sulfur and phosphorus groups, and induce reactive oxygen species and free radicals) [9,10]. This ensures AgNPs can target a broad range of bacteria while reducing their chance to develop resistance [11]. Recent studies showed additive and synergistic antibacterial effects of AgNPs combined with antibiotics against both Gram-negative and Gram-positive bacteria [12,13,14].

The physicochemical properties of AgNPs, such as size, shape, and surface chemistry, significantly affect their antibacterial activity. Small and medium-sized AgNPs (around 10 nm) show higher bactericidal activity than larger (>20 nm) NPs [15]. This effect has been ascribed to the relatively large surface area, the easier penetration of bacterial cell walls, and the ability to reach the nucleus. Although several methodologies for the fabrication of AgNPs have been described so far (i.e., chemical, physical, and green methods) [15,16,17], researchers are often dealing with issues related to the aggregation of AgNPs and thus the colloidal stability of their suspensions. Therefore, the identification of robust methodologies that allow accessing to small AgNPs while avoiding nanoparticle aggregation is a sought-after goal.

To address this challenge, organic and inorganic supports have been proposed as scaffolds for the preparation of AgNPs to improve the AgNPs’ stability and dispersibility [18]. Depending on the type of support, this strategy allows researchers to provide additional properties and to scale down the dose of AgNPs required for the bactericidal activity, thereby reducing side effects. It has been demonstrated that by tuning the composition of AgNP-bearing composite nanomaterials, it is possible to trigger different mechanisms for the antibacterial effects, thus preventing bacterial resistance [9,10].

In this context, the synthesis of three different AgNP-bearing composites (Figure 1) is described, using both organic and inorganic supports with intrinsic antibacterial properties.

Specifically, the commercially available sulfated cellulose nanocrystal (CNC, see ESI), functionalized reduced graphene oxide rGO-SA [19], and the commercially available titanium dioxide TiO_2_ were selected as organic (CNC, rGO-SA) and inorganic supports. Sulfated CNC is a biocompatible and cheap nanomaterial that can be easily obtained from renewable sources [20,21]. It is an excellent scaffold for the preparation of hybrid composites with inorganic nanoparticles, and nanocellulose-based antimicrobial materials have been investigated in several areas of application [22,23]. The synthesis of the functionalized graphene-based material rGO-SA was recently described by some of us [19]. This material consists of salicylic acid residues grafted on the surface of reduced graphene oxide sheets [19]. rGO-SA embedded onto a sample of cotton fabric efficiently provided a significant antimicrobial activity. TiO_2_ nanoparticles are produced in two forms: anatase and rutile. Their ability to interact with UV light and to produce reactive oxygen species has been widely investigated for the preparation of composite materials with a remarkable antibacterial effect against antibiotic-resistant bacteria [24].

In this work, the preparation and chemical/physical characterization of the CNC-AgNP, rGO-SA-AgNP, and TiO_2_-AgNP composites are described (Figure 1). In particular, the tuning of the AgNPs’ shape and size according to the type of bioactive support was investigated. Eventually, the antimicrobial activity of the new composites versus clinically relevant multi-drug-resistant bacteria and the fungus *Candida albicans* was assessed. These bacteria can “escape” the action of antibiotics through genetic mutations and the acquisition of mobile genetic elements, and together they represent new paradigms in pathogenesis, transmission, and resistance. In 2008, Rice introduced the acronym ESKAPE (i.e., ***E****nterococcus faecium*, ***S****taphylococcus aureus*, ***K****lebsiella pneumoniae*, ***A****cinetobacter baumannii*, ***P****seudomonas aeruginosa*, and ***E****nterobacter* species, including *Escherichia coli*) to identify those pathogens that are among the most common causes of life-threatening infections acquired in health facilities [25].

## 2. Materials and Methods

All reagents and solvents were purchased from Sigma-Aldrich (St. Louis, MO, USA), and they were used without any further purification if not specified otherwise. Cellulose nanocrystal was purchased from Celluforce (Windsor, QC, Canada). Graphene oxide was purchased from Nanesa (Roma, Italy). Titanium dioxide was purchased from Sigma-Aldrich as a mixture of the anatase and rutile forms (718467 Aldrich). UV-Vis spectra were recorded using a Varian Cary 4000 UV-Vis spectrophotometer using a 1 cm quartz cell.

### 2.1. X-ray Diffraction (XRD)

Powder X-ray diffraction (XRD) measurements were carried out on loosely packed powdered samples with a Bruker New D8 ADVANCE ECO diffractometer, equipped with a Cu Kα radiation source (1.5406 Å) and operating in θ-θ Bragg–Brentano geometry at 40 kV and 40 mA. The measurements were carried out in the range 30–90°, with a step size of 0.03° and collection time of 1 s.

### 2.2. Transmission Electron Microscopy (TEM)

Samples for TEM were dispersed in deionized water, placed in an ultrasound bath for 60 min, and dropped onto a lacey-carbon copper TEM grid. The sample TiO_2_-AgNPs were gently smashed in an agate mortar. The resulting powder was dispersed in isopropyl alcohol. It was placed in an ultrasound bath for 20 min and dropped onto a lacey-carbon copper TEM grid. The grids were analyzed after overnight drying with a ZEISS LIBRA200FE electron microscope.

### 2.3. Inductively Coupled Plasma Atomic Emission Spectroscopy (ICP-AES)

Inductively coupled plasma atomic emission spectroscopy (ICP-AES) was used to determine the concentrations of silver (Ag) and titanium (Ti); this was performed in triplicate using a Varian 720-ES inductively coupled plasma atomic emission spectrometer (ICP-AES). An accurately weighed amount of each sample was treated with microwave-assisted digestion (CEM MARS Xpress) using 1.0 mL of suprapure HNO_3_ obtained by sub-boiling distillation and 1.0 mL of suprapure H_2_O_2_. Each sample was thus diluted to 10 mL with ultrapure water (UHQ), spiked with 0.5 ppm of Ge used as an internal standard, and analyzed. Calibration standards were prepared using gravimetric serial dilution from commercial stock standard solutions of each element at 1000 mg L^−1^. The element determination was based on lines with wavelengths of 328.068 nm for Ag, 334.941 nm for Ti, and 209.426 nm for Ge. The operating conditions were optimized to obtain the maximum signal intensity, and a rinse solution constituting 2% *v*/*v* HNO_3_ was used between each sample to avoid memory effects.

### 2.4. Synthesis of CNC-AgNPs

To a stirred suspension of CNC (45.0 mg) in water (300 mL), NaBH_4_ (17 mg, 0.45 mmol) was added. Then, a solution of AgNO_3_ (76 mg, 0.45 mmol) in water (150 mL) was added dropwise to obtain a final 1 mM solution of AgNO_3_, 1 mM NaBH_4_, and 0.01% *w*/*v* of CNC. Then, the pH of the reaction medium was adjusted to 11 with a solution of NaOH (0.5 M in water), and the solution was stirred for 90 min at r.t. Then, the reaction mixture was filtered (PTFE filters, 0.4 µm) to yield 93 mg of a black/brown solid. UV-Vis λ_max_ = 392 nm. XRD: the peaks found at 2θ were 38.18°, 44.25°, 64.72°, 77.40°, and 81.55°, which are attributed to 111, 200, 220, 311, and 222 of the crystallographic planes of cubic Ag, respectively.

### 2.5. Synthesis of rGO-SA-AgNPs

To a stirred suspension of CNC (5.0 mg) and rGO-SA [19] (25 mg) in water (50 mL), NaBH_4_ (3.8 mg, 0.1 mmol) was added. Then, a suspension of CNC (5.0 mg) and AgNO_3_ (17 mg, 0.1 mol) in water (50 mL) was added to obtain a final 1 mM solution of AgNO_3_, 1 mM NaBH_4_, 0.25 mg/mL rGO-SA, and 0.01% *w*/*v* of CNC. Then, the pH of the reaction medium was adjusted to 11 with a solution of NaOH (0.5 M in water), and the solution was stirred for 90 min at r.t. The reaction mixture was then filtered (PTFE filters 0.4 µm) to yield 32 mg of a black/brown solid. The solid was resuspended in water with the aid of an ultrasound bath for 2 h (40 Hz) for further characterization. UV-Vis: λ_max_ = 404 nm (AgNPs) and λ_max_ = 264 nm (r-GO-SA). XRD: the peaks found at 2θ were 38.16°, 44.38°, 64.52°, 77.43°, and 81.52°, which are attributed to 111, 200, 220, 311, and 222 of the crystallographic planes of cubic Ag, respectively. 

### 2.6. Synthesis of TiO_2_-AgNPs

To a stirred suspension of TiO_2_ (800 mg) and a solution of NaBH_4_ (38 mg, 1 mmol) in water (650 mL), AgNO_3_ (170 mg, 1 mmol) in water (350 mL) was added dropwise to obtain a solution of 1 mM AgNO_3_, 1 mM NaBH_4_, and 0.8 mg/mL dispersion of TiO_2_. Then, the pH of the reaction medium was adjusted to 11 with a solution of NaOH (0.5 M in water), and the solution was stirred for 90 min at r.t. The solid was precipitated (8000 rpm, 10 min at 15 °C), the supernatant was removed, and the solid resuspended in milliQ water (250 mL); this process was repeated twice. The precipitate was then suspended in milliQ water (100 mL) and freeze-dried to afford 1.63 g of TiO_2_-AgNPs. The solid was resuspended in water with the aid of an ultrasound bath for 2 h (40 Hz) for further characterization. UV-Vis: λ = 388 nm (AgNPs) and λ_max_ = 296 nm and λ = 320 nm (TiO_2_). XRD: the peaks found at 2θ were 38.09°, 44.23°, 64.53°, 77.51°, and 82.85°, which are attributed to 111, 200, 220, 311, and 222, respectively, of the crystallographic planes of cubic Ag; 36.13°, 41.28°, 56.65°, and 68.89°, which are attributed to 101, 111, 220, and 301, respectively, of the crystallographic planes of tetragonal rutile TiO_2_; and 48.05°, 53.94°, 55.08°, 62.72°, and 70.34°, which are attributed to 200, 105, 211, 118, and 220, respectively, of the crystallographic planes of tetragonal anatase TiO_2_.

### 2.7. Microbial Strains, Media, and Culture Conditions

The antimicrobial activity of the AgNP hybrids was determined against *Klebsiella pneumonia*, *Escherichia coli*, *Pseudomonas aeruginosa*, *Staphylococcus aureus*, and *Candida albicans* with known resistance profiles. Specifically, *K. pneumoniae* with carbapenemase-producing (KPC) activity, extended-spectrum beta-lactamase (ESBL) *Escherichia coli*, multi-drug-resistant (MDR) *Pseudomonas aeruginosa*, and methicillin-resistant *Staphylococcus aureus* (MRSA) were used. All the isolates derived from positive blood cultures were retrieved from frozen glycerol stocks. Bacterial strains were streaked on fresh Trypticase soy agar plates with 5% sheep blood plate (bioMerieux) and sub-cultured to provide fresh colonies. *C. albicans* was cultured on selective Candida bromocresol green (BCG) agar medium.

### 2.8. MCC Evaluation

Minimal cytocidal concentrations (MCCs) were carried out according to the European Committee on Antimicrobial Susceptibility Testing (EUCAST) international guidelines [26]. Mueller–Hinton broth was used for the bacterial strains, while Sabouraud broth was used for the *Candida albicans*. A microbial suspension of 0.5 MacFarland was diluted 1:100 in broth medium and incubated in 96-well plates containing nanocomposites at concentrations ranging from 32 to 0.123 µg/mL, using a dilution factor of 1/2. Tests were performed in order to have 2.5 × 10^5^ cells/mL of bacteria and 2.5 × 10^4^ CFU/mL of yeasts in each well. After the incubation time at 37 °C for 24 h, MCC was defined as the lowest concentration that results in the death of 99.9% or more of the initial inoculum. Cell viability of the bacteria and yeasts was measured by seeding an aliquot of the contents of each well of the 96-well plate onto agar medium. Each test was performed in triplicate, and the experiments were repeated twice. Negative and positive controls were always set up. The negative control was represented by the culture medium only, while the positive control consisted of no treated cells (growth control).

### 2.9. MTS Cytotoxicity Test

The in vitro cytotoxicity of the tested composites was assessed with an MTS assay kit (Abcam, Cambridge, UK) on kidney epithelial cells from monkeys (VERO cells). VERO cells were grown in Dulbecco’s modified Eagle medium (DMEM) (Gibco, Life Technologies, Paisley, UK) supplemented with 10% heat-inactivated fetal bovine serum (Gibco, Life Technologies, UK), 1 mM glutamine (Gibco, Life Technologies, UK), 1% amphotericin B (Euroclone©, Milan, Italy), and 1% penicillin and streptomycin solution (Gibco, Life Technologies, UK). Cells were cultured at 37 °C with 5% CO_2_ and seeded in 96-well tissue culture plates at a concentration of 5 × 10^4^ cells/well. Then, CNC-AgNP, rGO-SA-AgNP, and TiO_2_-AgNP composites were added at 0.25×, 0.5×, 1×, 2×, and 4× the MCC value concentrations in DMEM, supplemented as previously described. The cells were incubated for 24 h and an MTS assay was conducted according to the manufacturer’s instructions. Briefly, 20 µL of MTS reagent were added to each well and the cells were incubated for 1 h; then, the absorbance was read at 490 nm. Viability was calculated as a percentage of the control.

## 3. Results

### 3.1. Preparation of CNC-AgNPs, rGO-SA-AgNPs, and TiO_2_-AgNPs

Sulfated CNC was used as a template for the preparation of the AgNPs by modifying a previously reported protocol [27,28,29].

Accordingly, a CNC suspension (0.15 mg/mL in H_2_O, Figure 1) was treated with sodium borohydride (3 mM) as a reductant, and then a solution of silver nitrate (1.5 mM) was added dropwise under vigorous stirring; then, the pH was adjusted to 11 with a NaOH solution (0.5 M in H_2_O). The reaction was monitored over time (0–90 min) using UV-Vis spectroscopy, and a progressive increase in the intensity of a peak at 392 nm (ESI, Appendix A) was observed. It was previously demonstrated that CNC stabilizes the dispersion of graphene materials [17]. Accordingly, in a second set of experiments, rGO-SA was also added to the dispersion of CNC (Figure 1, see ESI); then, the dispersion was treated in the same experimental conditions reported above. Finally, in a similar approach, TiO_2_ nanoparticles were used as a template for the preparation of AgNPs. Accordingly, a water dispersion of TiO_2_ (1.23 mg/mL) was treated with a solution of sodium borohydride (2.86 mM) and a solution of silver nitrate (1.54 mM); then, the pH was adjusted to 11 with a NaOH solution (0.5 M in H_2_O). The resulting AgNP nano-hybrids were fully characterized using UV-Vis spectroscopy, transmission electron microscopy (TEM) analysis, and X-ray diffraction (XRD).

### 3.2. Antimicrobial Activity

The different nanomaterials dissolved in solution showed an intrinsic coloring, preventing us from performing the minimum inhibitory concentration (MIC), which is evaluated by eye or with spectrophotometric reading. For this reason, we evaluated the cytocidal activity of the compounds by seeding an aliquot of the contents of all the wells of the 96-well plate on an agar medium. The minimal cytocidal concentration (MCC) of all the composites against the different microbial strains are summarized in Table 1.

Remarkably, all the nanomaterials showed MCC values of 2 µg/mL, while the TiO_2_-AgNPs sample showed an even lower MCC of 1 µg/mL. These results demonstrate how the AgNP-based formulations possess, at extremely low concentrations, broad-spectrum antimicrobial properties.

### 3.3. Cytotoxicity Assay

An MTS cytotoxicity assay conducted on the VERO cells shows in vitro cytocompatibility of all the three composites. The viability percentage compared with the control of the NCC-AgNPs and TiO_2_-AgNPs appears slightly superior to the control, indicating a proliferative effect. Instead, the rGO-SA-AgNPs composite does not show a proliferative effect and seems to have a slight decrease in viability compared with the control, but none of the tested concentrations have a median viability below 70%, which is commonly a symptom of no cytotoxic effect (Figure 2).

## 4. Discussion

Some protocols have been reported so far for the preparation of CNC-AgNPs and TiO_2_-AgNPs [24,27,28,29,30]; however, a study that compares the effect of the template in the same experimental conditions is missing. In particular, CNC works as both a capping and dispersing agent and allows AgNP dispersions with improved colloidal stability [29]. Conversely, the synthesis of the rGO-SA-AgNPs, using the rGO-SA as a template, is reported for the first time in this work. Specifically, organic (CNC and rGO-SA) and inorganic (TiO_2_) supports were treated in the same experimental conditions; thus, the effect of the support on the AgNPs’ size and shape was evaluated. The UV-Vis spectrum of the dispersion of the CNC-AgNP hybrids confirms the presence of the characteristic surface plasmon resonance (SPR) absorption band at 392 nm. The shape of the peak suggests a narrow distribution of the AgNPs’ size (ESI, Appendix A). The UV-Vis spectrum of the rGO-SA-AgNPs shows the SPR absorption of the AgNPs at 395 nm and a second broad peak at 264 nm associated with the rGO-SA (Appendix A), as previously reported [17]. In the UV-Vis spectrum of the TiO_2_-AgNPs, the SPR peak of the AgNPs (λ_max_ = 389 nm) is overlapped with the broad absorption of the TiO_2_ (Appendix A). 

TEM images (Figure 3) were carried out to evaluate the morphology of the prepared AgNP-bearing hybrids. According to the UV-Vis spectra, large aggregates were not detected across all the samples; rather, particles well-shaped, crystalline, and dispersed in the templates were observed. Notably, mean particle size changes, according to the different templates used, were observed, with a nanoparticle size distribution < 15 nm. In most of the cases, bigger nanoparticles were a small fraction of the poly-dispersion. In particular, the AgNPs in the CNC-AgNPs hybrid (Figure 3A) showed a size around 13.90 nm (Appendix A, Appendix A) with a slightly asymmetric distribution ranging from 2 to 40 nm, with 75% of the nanoparticles < 16 nm. The rGO-SA-AgNPs hybrid (Figure 3B, Appendix A) showed the formation of smaller nanoparticles compared with the CNC-AgNPs sample, with a mean size centered at 4.9 nm. Although with a broad tail ranging to 20 nm, 90% of the nanoparticles showed a size < 10 nm (Appendix A, ESI). The rGO-SA was in the form of few-layered graphene with a homogeneous distribution of AgNPs on its surface. TEM images of the TiO_2_-AgNPs sample (Figure 3C) showed a narrow distribution of AgNPs with a mean size of 2.2 nm (Appendix A, Appendix A, ESI), with 90% of the particles < 3 nm.

The XRD patterns of the CNC-AgNPs and rGO-SA-AgNPs (Figure 4 and Appendix A) show all the reflexes belonging to the $Fm\bar{3}m$ (face-centered cubic structure) characteristic of the silver space group of the Ag crystal structure in addition to the cubic structure of silver; the presence of a satellite crystallographic phase, identified as CNC, indicates that the AgNPs are effectively embedded in a cellulose matrix. In the TiO_2_-AgNPs sample, the XRD pattern show beside the Ag bands the reflexes belonging to the P42/mnm and I41/amd (tetragonal structure) space group of the TiO_2_ anatase and rutile forms, respectively. The fitting process, using the Pawley method and thus refining only the lattice parameters (*a*) and the peak width, was carried out on the whole series and the results are reported in Appendix A. The as-calculated lattice parameters are in good accordance with the one reported in the literature for AgNPs obtained with the same synthetic method [JCPDS 89-3722] and TiO_2_ nanoparticles [PDF 89-0554 for rutile and PDF 04-0477 for anatase]. The crystal size also shows a good agreement with the size evaluated by TEM statistical analysis, corroborating the synthesis of monocrystal nanoparticles.

Accordingly, all the selected templates provided spherical-shaped nanoparticles homogeneously distributed in the template. Then, a nanoparticle size distribution < 15 nm was observed with smaller AgNPs (around 2.2 nm) using the TiO_2_ as template. Finally, the amount of silver in the composites was analyzed with ion-coupled plasma atom emission spectroscopy (ICP-AES), which resulted in a roughly full recovery of the silver used in the reaction (ESI, Appendix A).

The three hybrid composites synthetized were tested for their antimicrobial activity. We tested one Gram-positive, *S.aureus*, a major human pathogen; three Gram-negative bacteria that are frequently isolated from nosocomial infections, *K. pneumoniae*, *E. coli,* and *P. aeruginosa*; and the predominant fungal pathogen, *C. albicans*. All tested strains were previously assessed for sensitivity to commonly used antimicrobial drugs in clinical practice through broth microdilution assays, and therefore categorized as sensitive or resistant according to the criteria established by EUCAST. The nanocomposites exhibited very low cytotoxic concentrations against all the microorganisms tested. Moreover, their efficacy is confirmed against the yeast *C. albicans* and both Gram-negative and Gram-positive bacteria, showing a wide spectrum of applications in a clinical setting. The toxicity against mammalian cells was also explored; the vitality test showed no cytotoxic effect against a human fibroblast cell line (Figure 4).

## 5. Conclusions

In this work, the use of three different templates, both organic and inorganic, for the preparation of AgNP hybrids was studied. Specifically, CNC, rGO-SA, and TiO_2_ were revealed as suitable templates for the fine-tuning of AgNP size, and allowed the preparation of small-sized AgNPs. Moreover, the synergic action of the intrinsic antimicrobial properties of the selected templates and the AgNPs provided the hybrid nanomaterials with remarkable antimicrobial activity. This activity was remarkable against all the multi-drug-resistant bacteria and yeasts tested in this study, which are the leading cause of nosocomial infections worldwide [31]. These findings pave the way for the development of effective nanomaterials that may be used for further applications such as antimicrobial coatings.

## Data Availability

Data available on request to the corresponding authors.

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
