# Peer review of "Shaping Silver Nanoparticles’ Size through the Carrier Composition: Synthesis and Antimicrobial Activity"

_nanomaterials, 2023, doi:10.3390/nano13101585_

Round 1

Reviewer 1 Report

Article topic - Shaping silver nanoparticles size through the carrier composition: synthesis and antimicrobial activity

Comments & Suggestions

Abstract

* There is no proper mention of the results or conclusion in the abstract. It must be included. Big portion of the abstract is allocated for background information.

Lines 20-23 = The sentence is too long. It distracts the meaning. Please break the sentence.

Line 26,30 – Avoid using “we”.

Line 26 – “AgNPs hybrids” needs to be corrected to AgNP hybrids.

Line 30, 31– Check the grammar of “AgNPs shape”. I think it should be “AgNPs’ shape”. This applies to the whole article.

Line 30 – Avoid using “then”.

Please revise the abstract.

Introduction

* Overall, the flow of the introduction must be improved. The aims and objectives are not clearly presented in the text. The novelty of the study should be explained more. Revise the introduction.

Line 40 – at the end of the sentence, add “and”. “food packaging and clothing.”

Line 45-46 = Poor sentence structure. Please revise.

Line 56, 60, 92 – The full stop should be put after the reference. Not before the reference. This applies to the whole document.

Line 57 – Reference style of 12-14  is different here. Why?

Line 60 - bigger sized NPs ? Please indicate the size of NPs or what you mean by “bigger”.

Line 73 - Avoid using “we”. This applies to the whole document.

Line 74- Synthesis of AgNPs using both organic and inorganic supports; is this a novel method or has this method been used in previous literature?

* There is no mention of abbreviations. Please indicate the abbreviation where relevant. This applies to the whole document.

Line 84,90,91 – Poor sentence structure, please check grammar.

Line 99 – Please break the sentence, it is too long.

Line 101 – Include names of the pathogen next to the word ESKAPE.

Line 104 – Add the reference at the end of the sentence.

Methodology

Line 109 – “anatase e rutile”. Check spellings?

Line 120-122 – Sentence is too long.

Line 134- 135 – The sentence is grammatically incorrect. Please revise.

Line 178- 180 – The language structure is poor and grammatically incorrect. Please revise.

Line 184 – “cultured” is more appropriate than “cultivated” since this is dealing with microbiological cultures.

Line 185 – What is the reason to use the term MCC? Why it is not MIC (Minimum inhibitory concentration)?. Please specify.

Line 188 – What is reason to choose Muller Hinton broth for the bacterial strains? Spell check please.

Line 191- Concentration range for the nanocomposites in 96 multi-well plate is 32 to 0.123 μg/mL. What are the other different concentrations used in between this range? Specify.

Line 192 – The incubation time is missing. Indicate that. Did you use the same cell concentration (2.5 × 191 105 cells/ml) for the yeast strain (Candida albicans)? Why?

Line 193 – What is the method employed to detect the cells which are dead 99%? Explain the methodology for both bacterial cultures and for the yeast strain.

Line 194 – What does it mean by “and in different days”. Describe it meaningfully. Mention the positive and the negative controls.

Results

* Overall, the results are not cited enough with the previous studies. They should be compared with other studies in the literature and cited properly.

Line 198 – Reference pattern is different again. Revise.

* Figure 2 is missing. Revise the figures and numbers.

Line 203- Poor sentence structure for the title of scheme 1. Revise. Is scheme 1 same as Figure 2?

Line 269 – Methodology 3.1 Is this numbering correct? Because, before this subtitle, lot of results have been described.

Line 277 – Strain name should be in italic form.

Conclusion

The sentences are too long. Please break the sentences in a more meaningful way. Results and findings of the study are not directly/ properly concluded. Mention the future directions. Mention the limitations of the study.

many typos are in the manuscript. Check spellings. Sentences are too long

Author Response

Point by point response to the reviewers’ comments on Manuscript ID: nanomaterials-2345459.

Reviewer 1:

1)      There is no proper mention of the results or conclusion in the abstract. It must be included. Big portion of the abstract is allocated for background information.

Response: The abstract has been modified accordingly removing two sentences about the state of the art and adding a new sentence concerning the results

2)      Lines 20-23 = The sentence is too long. It distracts the meaning. Please break the sentence.

Response: The long sentence was divided in two shorter sentences.

3)      Line 26,30 – Avoid using “we”.

Response: the text was amended accordingly

4)      Line 26 – “AgNPs hybrids” needs to be corrected to AgNP hybrids.

Response:  Corrected

5)      Line 30, 31– Check the grammar of “AgNPs shape”. I think it should be “AgNPs’ shape”. This applies to the whole article.

Response: Corrected

6)      Line 30 – Avoid using “then”.

Response: Substituted by “subsequently”

7)      * Overall, the flow of the introduction must be improved. The aims and objectives are not clearly presented in the text. The novelty of the study should be explained more. Revise the introduction.

Response: The text has been revised accordingly.

8)      Line 40 – at the end of the sentence, add “and”. “food packaging and clothing.”

Response: “And” was added

9)      Line 45-46 = Poor sentence structure. Please revise.

Response: The sentence was modified

10)     Line 56, 60, 92 – The full stop should be put after the reference. Not before the reference. This applies to the whole document.

Response: The manuscript has been modified accordingly.

11)     Line 57 – Reference style of 12-14 is different here. Why?

Response: We just intended to mentions all references between 12 and 14 (12, 13, 14)

12)     Line 60 - bigger sized NPs? Please indicate the size of NPs or what you mean by “bigger”.

Response: We modified the sentence to make it clearer

13)     Line 73 - Avoid using “we”. This applies to the whole document.

Response: Corrected

14)     Line 74- Synthesis of AgNPs using both organic and inorganic supports; is this a novel method or has this method been used in previous literature?

Response: References related to previously reported protocols have been included. In this work, both the organic and inorganic scaffold have been treated in the same experimental conditions. This approach allowed us to provide a comparative study and to unveil the effect of the template on AgNPs size and shape. A sentence has been included in the main text to clarify.

15)     * There is no mention of abbreviations. Please indicate the abbreviation where relevant. This applies to the whole document.

Response: Abbreviations have been included at the end of the main text

16)     Line 84,90,91 – Poor sentence structure, please check grammar.

Response: The sentence has been largely modified to make it easier to be read

17)     Line 99 – Please break the sentence, it is too long.

Response: The sentence was modified

18)     Line 101 – Include names of the pathogen next to the word ESKAPE.

Response: The sentence was modified accordingly

19)     Line 104 – Add the reference at the end of the sentence.

Response: The reference was moved to the end of the sentence

20)     Line 109 – “anatase e rutile”. Check spellings?

Response: “and” was inserted instead of “e”

21)     Line 120-122 – Sentence is too long.

Response: The sentence was divided in two

22)     Line 134- 135 – The sentence is grammatically incorrect. Please revise.

Response: The sentence was amended

23)     Line 178- 180 – The language structure is poor and grammatically incorrect. Please revise.

Response: The sentence was amended

24)     Line 184 – “cultured” is more appropriate than “cultivated” since this is dealing with microbiological cultures.

Response: the words were substituted

25)     Line 185 – What is the reason to use the term MCC? Why it is not MIC (Minimum inhibitory concentration)? Please specify.

Response: The failure to perform MICs has been explained and added in the results section. I reproduce the text below for your clarity:

-The different nanomaterials dissolved in solution showed an intrinsic coloring which did not allow us to perform the Minimum Inhibitory Concentration (MIC) which is evaluated eye-metrically or by spectrophotometric reading. For this reason, we evaluated the cytocidal activity by inoculating an aliquot of the contents of all the wells of the 96-multi-well plate on agar medium-

26)     Line 188 – What is reason to choose Muller Hinton broth for the bacterial strains? Spell check please.

Response: The choice of Mueller Hinton medium is dictated by the EUCAST guidelines for performing micro-broth dilution tests. Spell check has been controlled

27)     Line 191- Concentration range for the nanocomposites in 96 multi-well plate is 32 to 0.123 μg/mL. What are the other different concentrations used in between this range? Specify.

Response: We have clarified in the text the dilution factor used for compounds

28)     Line 192 – The incubation time is missing. Indicate that. Did you use the same cell concentration (2.5 × 191 105 cells/ml) for the yeast strain (Candida albicans)? Why?

Response: The incubation time has been indicated. The different concentrations of bacteria and yeasts was specified in the text

29)     Line 193 – What is the method employed to detect the cells which are dead 99%? Explain the methodology for both bacterial cultures and for the yeast strain.

Response: The method was specified as required

30)     Line 194 – What does it mean by “and in different days”. Describe it meaningfully. Mention the positive and the negative controls.

Response: The requested clarifications have been added to the text

31)     * Overall, the results are not cited enough with the previous studies. They should be compared with other studies in the literature and cited properly.

Response: References of previously reported studies have been included in the text. Please check response 14)

32)     Line 198 – Reference pattern is different again. Revise.

Response: The reference has been revised

33)     Figure 2 is missing. Revise the figures and numbers.

Response: the numbering of each figure was revised

34)     Line 203- Poor sentence structure for the title of scheme 1. Revise. Is scheme 1 same as Figure 2?

Response: The caption was modified. The numbering of figures was modified

35)     Line 269 – Methodology 3.1 Is this numbering correct? Because, before this subtitle, lot of results have been described.

Response: The text has been revised accordingly

36)     Line 277 – Strain name should be in italic form.

Response: corrected

36)     The sentences are too long. Please break the sentences in a more meaningful way. Results and findings of the study are not directly/ properly concluded. Mention the future directions. Mention the limitations of the study.

Response: The text was revised accordingly

Reviewer 2 Report

Comments to the Authors:

In this manuscript authors investigated investigate the tuning of the AgNPs shape and size according to the type of bioactive support and the antimicrobial activity of the AgNPs composites was assessed on clinically relevant multi-drug-resistance bacteria and the fungus Candida albicans. However, the paper needs minor improvement before acceptance for publication. My detailed comments are as follow:

1.      In the introduction section authors should introduce following relevant articles related to AgNPs properties

a.      doi.org/10.1007/s11164-020-04165-0

b.      doi.org/10.1007/s40089-021-00362-w and also others.

2.      There are some typos and grammatical errors.

3.      Authors should histograms of AgNPs size obtained from microscopic images.

4.      Marked the peaks in the XRD pattern with name.

5.      The writing of conclusion section should be improved.

6.      The quality of the Figure 4 should be improved.

Minor editing is required.

Author Response

Point by point response to the reviewers’ comments on Manuscript ID: nanomaterials-2345459.

Reviewer 2:

1)      In the introduction section authors should introduce following relevant articles related to AgNPs properties

  1. doi.org/10.1007/s11164-020-04165-0
  2. doi.org/10.1007/s40089-021-00362-w and also others.

Response: References were included

2)      There are some typos and grammatical errors.

Response: We made our best to amend the text

3)      Authors should histograms of AgNPs size obtained from microscopic images.

Response: Histograms are included in the ESI.

4)      Marked the peaks in the XRD pattern with name.

Response: Figure 3 was revised

5)      The writing of conclusion section should be improved.

Response: The conclusion section was modified

6)      The quality of the Figure 4 should be improved.

Response: The quality of Figure 4 (now Figure 3) was improved

Round 2

Reviewer 1 Report

Article topic - Shaping silver nanoparticles size through the carrier composition: synthesis and antimicrobial activity

Abstract

Abstract yet to be revised.

(1)    Line 27 = “TiO2” should be abbreviated in the first place it appears on the paper.

(2)    Line 24-25 =  Sentence is not broken into short sentences. Only a “comma” was inserted instead. Please revise.

(3)    Line 29-31 – Still a poor sentence structure. Revise. “formation of nanosized Ag nanoparticles”- Avoid repetition. Revise.

* Add a direct conclusive sentence at the end of the abstract.

Introduction

(4)   Line 48,52,55 = Again two different reference styles are detected. It was not corrected from the last time. If the reference means 7 and 8, it should be written [7-8]. Check with the journal guidelines and revise. This applies to the whole manuscript.

(5)   Line 80 = Font size is different in the citation. Revise.

Methodology

(6)   Line 121-123 = The sentence is too long, avoid using commas and elongating the sentences. It is unnecessary. “The resulting powder was dispersed in isopropyl alcohol, placed in an ultrasound bath for 20 minutes, and dropped onto a lacey-carbo Copper TEM grid ” can be well-written as “The resulting powder was dispersed in isopropyl alcohol. It was placed in an ultrasound bath for 20 minutes and dropped onto a lacey-carbo Copper TEM grid.”

(7)   Line 131-134 = Sentence is too long. Revise.

(8)   Line 136 = Spell check.

(9)   Line 137-138 = Poor sentence structure. Revise.

(10) Line 144 = Abbreviate “r.t”. (Room temperature)

* Overall, the methodology section should be revised to avoid “then” in each time it appears unnecessarily. Adopt appropriate scientific writing style.

(11)   Line 184 = Break the long sentence.

(12)   Line 189 = Ref. font is changed.

(13)   Line 194 = “2.5 x 104 of yeasts”- Isn’t this a cell concentration? If so, add the units. Revise.

(14)   Line 198 = What does it mean by “repeated in different days”? Revise the sentence in a meaningful way. “Negative and positive controls were always included.” can be changed to “Negative and positive controls were set up.” (preferably)

(15)   Line 199-201 = sentence is too long. It confuses the meaning. Revise

(16)   Line 208= Which antibiotic did you use? Penicillin or Streptomycin? Be specific.

(17)   Line 210= Check superscript and subscript. This applies to the whole document.

Results

(18)   Line 245 = “AgNPs nano-hybrids” needs to be corrected to AgNP nano- hybrids. This comment was given in the first round. But it was not corrected in every place. Revise.

(19)   Line 293 – 294 = ICP-AES abbreviation is completely wrong here. Avoid this mistake.

(20)   Line 305 = It should be “different bacterial strains”

(21)   Line 331-332 = I what way these findings pave the way to develop effective nanomaterials? Please specify the sentence.

* Overall, the results can be more-well organized and presented. The way of presenting results is not clear enough. It should be improved. Language editing should be applied to the whole document. English writing style need to be improved. Mention the future directions. Mention the limitations of the study.

* The manuscript is missing  the discussion section. Why is that? According to nanomaterials-MDPI  template- Discussion section should be included. It should be detailed and cited with appropriate references. The results of this study should be compared with previous research. Without the discussion part, this manuscript is not worth publishing unless the new nanomaterials-MDPI  template doesn’t require a discussion section.

Author Response

Reviewer 1:

Abstract yet to be revised.

  • Line 27 = “TiO2” should be abbreviated in the first place it appears on the paper.

Response: Text has been changed accordingly

  • Line 24-25 =  Sentence is not broken into short sentences. Only a “comma” was inserted instead. Please revise.

Response: Text has been changed accordingly

  • Line 29-31 – Still a poor sentence structure. Revise. “formation of nanosized Ag nanoparticles”- Avoid repetition. Revise.

Response: Sentence has been revised as you suggested

* Add a direct conclusive sentence at the end of the abstract.

Introduction

  • Line 48,52,55 = Again two different reference styles are detected. It was not corrected from the last time. If the reference means 7 and 8, it should be written [7-8]. Check with the journal guidelines and revise. This applies to the whole manuscript.

Response: References styles has been corrected all over the manuscript

  • Line 80 = Font size is different in the citation. Revise.

Response: Done

Methodology

  • Line 121-123 = The sentence is too long, avoid using commas and elongating the sentences. It is unnecessary. “The resulting powder was dispersed in isopropyl alcohol, placed in an ultrasound bath for 20 minutes, and dropped onto a lacey-carbo Copper TEM grid ” can be well-written as “The resulting powder was dispersed in isopropyl alcohol. It was placed in an ultrasound bath for 20 minutes and dropped onto a lacey-carbo Copper TEM grid.”

Response: The sentence has been changed as you suggested

  • Line 131-134 = Sentence is too long. Revise.

Response: In our opinion the two sentences are clear and not excessively long

  • Line 136 = Spell check.

Response: Done

  • Line 137-138 = Poor sentence structure. Revise.

Response: Revised

  • Line 144 = Abbreviate “r.t”. (Room temperature)

Response: Done

* Overall, the methodology section should be revised to avoid “then” in each time it appears unnecessarily. Adopt appropriate scientific writing style.

(11)   Line 184 = Break the long sentence.

Response: The sentence is already divided in two parts.

  • Line 189 = Ref. font is changed.

Response: Done

  • Line 194 = “2.5 x 104of yeasts”- Isn’t this a cell concentration? If so, add the units. Revise.

Response: Done

  • Line 198 = What does it mean by “repeated in different days”? Revise the sentence in a meaningful way. “Negative and positive controls were always included.” can be changed to “Negative and positive controls were set up.” (preferably)

Response: The sentence has been changed as you suggested

  • Line 199-201 = sentence is too long. It confuses the meaning. Revise

Response: Revised

  • Line 208= Which antibiotic did you use? Penicillin or Streptomycin? Be specific.

Response: Both antibiotics were used. We modifies the sentence

  • Line 210= Check superscript and subscript. This applies to the whole document.

Response: Done

Results:

  • Line 245 = “AgNPs nano-hybrids” needs to be corrected to AgNP nano- hybrids. This comment was given in the first round. But it was not corrected in every place. Revise.

Response: AgNPs nano- hybrids was corrected all over the manuscript

  • Line 293 – 294 = ICP-AES abbreviation is completely wrong here. Avoid this mistake.

Response: The abbreviation is completely correct

  • Line 305 = It should be “different bacterial strains”

Response: the presence of a yeast implies that we cannot use bacteria but microbial

  • Line 331-332 = I what way these findings pave the way to develop effective nanomaterials? Please specify the sentence.

Response: The authors intend that the promising results (antimicrobial efficacy and absence of cytotoxicity) lay the foundations for the possible future use of these nanomaterials in the clinical setting

* Overall, the results can be more-well organized and presented. The way of presenting results is not clear enough. It should be improved. Language editing should be applied to the whole document. English writing style need to be improved. Mention the future directions. Mention the limitations of the study.

* The manuscript is missing  the discussion section. Why is that? According to nanomaterials-MDPI  template- Discussion section should be included. It should be detailed and cited with appropriate references. The results of this study should be compared with previous research. Without the discussion part, this manuscript is not worth publishing unless the new nanomaterials-MDPI  template doesn’t require a discussion section.

Round 3

Reviewer 1 Report

Authors have improved the manuscript.

Please use template

There are five sections Introduction, method, results and discussion and conclusion.

however, the discussion is serious missing. Please create the discusion section 4 discussion. Please discuss the results in the 4. Discussion. 

Please use Palatino Linotype number 10 in the main test. Font of figure legends should be number 9. 

CNC-AgNPs, rGO-SA-AgNPs and TiO2-AgNPs are no necessary to be bold letter.

line 325 to 331 Please use justified in the alignment in the paragraph to arrange  right margin.

Please arrange line 350. The letters are overlapping. 

Author Response

Reviewer comments

Authors have improved the manuscript.

Please use template

There are five sections Introduction, method, results and discussion and conclusion.

however, the discussion is serious missing. Please create the discusion section 4 discussion. Please discuss the results in the 4. Discussion. 

Response: We apologize to the reviewer for our omission. The discussion was incorrectly inserted together with the results. We proceeded to insert paragraph 4 with the discussion section in the main text.

Please use Palatino Linotype number 10 in the main test. Font of figure legends should be number 9. 

Response: We changed the figure legends accordingly.

CNC-AgNPs, rGO-SA-AgNPs and TiO2-AgNPs are no necessary to be bold letter.

Response: Bold letters have been removed

line 325 to 331 Please use justified in the alignment in the paragraph to arrange right margin.

Response: We justified the paragraph as suggested.

Please arrange line 350. The letters are overlapping. 

Response: Done